# Analysis and Mitigation of Pulse-Pile-Up Artifacts in Plasma Pulse-Height X-ray Spectra

**Taosif. Ahsan [1,2], Charles P. S. Swanson [2,3], Chris Galea [3], Sangeeta P. Vinoth [2,3], Tony Qian [2], Tal Rubin [2] and Samuel A. Cohen [2,\*]**

1 Physics Department, Princeton University, Princeton, NJ 08540, USA
2 Princeton Plasma Physics Laboratory, Princeton University, Princeton, NJ 08543, USA
3 Princeton Fusion Systems, Plainsboro, NJ 08536, USA
\* Correspondence: scohen@pppl.gov

**Abstract:** Pulse pile-up in pulse-height energy analyzers increases when the incident rate of pulses increases relative to the inverse of the dead time *per* pulse of the detection system. Changes in the observed energy distributions with incident rate and detector-electronics-formed pulse shape then occur. We focus on weak high energy tails in X-ray spectra, important for measurements on partially ionized, warm (50–500 eV average electron energy), pure hydrogen plasma. A first-principles two-photon pulse-pile-up model is derived specific to trapezoidal-shaped pulses; quantitative agreement is found between the measurements and the model's predictions. The model is then used to diagnose pulse-pile-up tail artifacts and mitigate them in relatively low count-rate spectra.

**Keywords:** X-ray; pulse pile up; energy analyzers; plasma; 2-photon; dead time; count rate

## 1. Introduction

Solid-state detectors operating in the pulse-height mode are used to measure energy distributions of incident radiation [1,2]. In these detectors, the amount of electrical charge released by the impact of a single photon is proportional to the energy of the photon. The energy resolution of these detectors is set by the timing and statistics of the generated and integrated charge, thermal noise, and the accuracy of the conversion of that charge to voltage [3]. In this paper, we will focus on energy analysis of X-rays emitted from plasmas. The method may also be applied to energy-resolving charged-particle detection.

Pulse-height X-ray detectors are used for applications where moderate resolution over a broad energy spectrum is more advantageous than high resolution in a narrow spectrum. The latter, for example, is obtainable with a crystal spectrometer [4]. The former is preferred for applications such as X-ray fluorescence spectroscopy of material samples [5] and electron temperature measurement *via* broad-spectrum Bremsstrahlung of warm or hot plasmas [6].

Experiments on the Princeton Field-Reversed-Configuration (PFRC-2) device explore nearly pure, *ca.* 99%, partially ionized, warm hydrogen plasmas [7]. For these, great interest lies in the tails of the X-ray spectrum. Small tails of high-energy electrons in the energy distribution (EED), even comprising less than 1% of the plasma density, can have large effects on resistivity, stability, and reaction rates of the plasma. Amptek X-123 Fast Silicon Drift Detector (SDD) pulse-height X-ray systems [5] have been used to detect and analyze X-rays emitted by electrons in PFRC-2 experiments. This paper focuses on spectral tails, a topic not encountered in the element-analysis application of SDDs, and represents an extended arena of their use.

Because the free charge generated in SDDs is $\sim 4.4 \times 10^{-20}$ C/eV of incident photon energy [8], the useful low energy limit of these detectors, based on resolution, is about 100 eV. These detectors are sensitive to lower-energy photons (VUV, UV, and visible), though these photons are not spectrally resolved.

For a high photon flux, more than one may arrive at the detector within the time that the free charge is integrated, the dead time *per* pulse, $t_d$, the minimum time between two pulses at which they can be recognized by the electronics as distinct. As a result, the detector interprets near-coincident impacts of multiple photons as a single photon liberating charge equal to the sum of the liberated charge from the multiple impacts. This is termed pulse pile-up (PPU). Unraveling the true photon energy distribution when PPU occurs requires knowledge of the specific shape of the pulse and the peak detection processes.

As a result of PPU, the measured energy spectrum is corrupted in several ways. Firstly, PPU may produce false peaks in the distribution, located at the sum of the energies of two peaks. ("Real" peaks in the distribution correspond to elemental line radiation from an X-ray tube target, gamma radiation of radionuclides, or photon scattering mechanisms, e.g., Compton scattering.) Secondly, there will be a decrease in X-ray counts at low energies. Thirdly, the location of a peak may be shifted to higher energy if abundant low energy photons pile up with photons of the higher energy peak. Finally, PPU may add a tail to the distribution, which, in the case of two-photon pile-up, could reach up to twice the maximal photon energy in the experiment. PPU may also corrupt an already existing tail by giving a higher count rate in the tail region than the true count rate. It is of paramount interest to distinguish between tails that are artifacts of PPU and tails that have a physical origin.

Activating manufacturer-supplied software that reduces PPU—software termed "pile-up rejection" (PUR)—proved ineffective at fully eliminating pile-up in many spectra in PFRC-2 experiments for reasons explained in Section 3. Thus, an analytical or algorithmic method was sought to analyze these spectra with the key objective to extract the electron energy distribution function (EEDF) from the X-ray energy distribution function (XEDF), primarily in the tail region.

The effect of PPU has been previously discussed by many authors. Datlowe analyzed the role of the shape of the pulse waveform and developed a method to calculate the effects efficiently [9], Guo et al. used a Monte-Carlo method, MCPUT, to correct the spectral distortion from PPU [10]. Taguchi et al. derived methods to correct the peak and tail pile-up effect for non-paralyzable detectors [11]. Wang et al. analyzed, for different pulse shapes, the effect of pulse pile-up on the spectrum for a double-sided silicon strip detector, accounting for the spatial distribution of photon interactions [12]. However, for digitized trapezoidal-shaped pulse-height detection systems, the effect of PPU on and the analytic mitigation of tail distortion in measured spectra have not previously been analyzed in detail.

This paper is a step towards understanding how PPU affects the tail region of spectra for detector-formed trapezoidal pulses. We focus on relatively low count rate ($\leq 0.1/$dead-time) spectra where primarily only two-photon pile-up needs to be considered. Extension of this work to multi-photon pile-up will be necessary to develop an analytical tool to diagnose and mitigate pile-up effects in the tail regions of higher count-rate spectra.

## 2. Sample Spectra with PPU

To study this immitigable PPU and test models, several experiments were performed. First, X-ray emission was measured from a graphite-target X-ray tube with incident electron beams at various currents and fixed electron energy: $E_e = 5$ keV for the cases presented here. Using a solid graphite target reduces poorly quantifiable VUV emission—attributed to hydrogen, carbon, nitrogen, and oxygen lines and low energy Bremsstrahlung in the PFRC-2 experiments to be described shortly—that generates PPU. Fortuitously, the truncated solid-target Bremsstrahlung spectrum mimics the EEDF predicted by Hamiltonian simulations of some FRCs [13], providing a means to evaluate these codes. For these measurements, the SDD's PUR feature was disabled.

In these studies, and in many SDDs, trapezoidal pulses have equal rise and fall times, $t_r$, and a short duration flat top, $t_f$. Amptek X-123 SDDs define $t_d = t_r + t_f$ [5].

Figure 1 shows spectra measured using higher (750 nA) and lower (190 nA) X-ray-tube electron currents, corresponding to higher (65,000 counts per second, 65 kcps) and lower

(14 kcps) X-ray count rates. (We use the terms count rate, CR, and $\mu$ interchangeably throughout the paper.). At lower count rate (blue), the spectrum has bright C, O, and Si K-$\alpha$ lines and solid-target Bremsstrahlung. The ratio of the 1740 eV Si peak (due to fluorescence from the SDD's C1 window) [5] to the signal at 5 keV is $10^4$. At higher count rate (red), the ratio has dropped to 1300, an indication of PPU. The PPU-generated (unphysical) tail above 5 keV has a near-exponential shape of slope twice that in the region $1400 < E < 4000$ eV. Though $\mu t_d$ is low even for the high CR case, *ca.* $4 \times 10^{-2}$, changes in the spectrum's unphysical tail, above 5 keV, are clear. (The solid angle of the SDD is small, hence negligible correlation of X-ray arrivals from a single electron impact on the graphite will occur.). The "predicted spectrum" (yellow) will be discussed in Section 7.

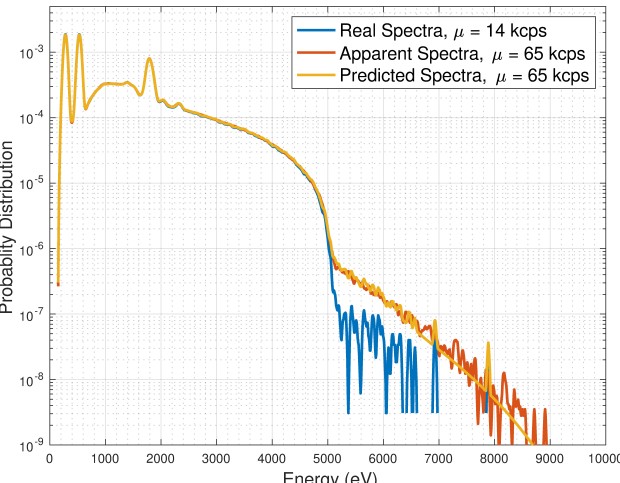

**Figure 1.** X-ray spectra at two count rates, 14 kcps (blue) and 65 kcps (red), for a 5-keV electron beam impacting a carbon target. Noise was reduced using a 50-eV-wide weighted moving-average filter. The slow-channel dead time was 212 ns. For comparison, the predicted spectrum (yellow) was derived using the two-photon PPU model, to be described in Section 7.

Second, measurements were performed on X-ray emission from seed plasmas in the PFRC-2 filled with hydrogen ($H_2$) gas to a center-cell pressure $153 \pm 4$ µTorr. The magnetic field at the PFRC-2 center was 197 Gauss and 550 W forward power was applied by a capacitively coupled rf source. The SDD, located 41 cm from the PFRC-2 major axis, viewed the plasma through an 82-mm$^2$-area aperture, 10.7 cm from the axis. $\mu t_d$ was changed by varying $t_d$; the count rate, $\mu$, was kept constant at 9.54 kcps. We measured spectra with two different $t_r$, 1.0 µs and 5.6 µs, and $t_f = 0.012$ and 0.2 µs, respectively. As shown in Figure 2, the region between 650 eV and 1100 eV has a 50% rise in count rate at the higher $\mu t_d$. For $E > 1100$ eV, the tail increased by a smaller percentage. The C, N, and O peaks shift to 17 eV higher energy while the CR from 200–510 eV falls 20%, see inset. All are indications of pile up.

Figures 1 and 2 support the claim that PPU compromises spectra. In the discussions thus far, the evidence for pile up in these spectra has been qualitative. In this paper, we present a two-photon PPU model of trapezoidal-shaped pulses that successfully explains the amplitude of the high-energy tail for low count-rate spectra. We provide an algorithmic/analytical way of deducing whether a tail is a complete artifact of PPU or the deformation of a true tail. Moreover, we provide an analytic means to recover true tails when PPU occurs.

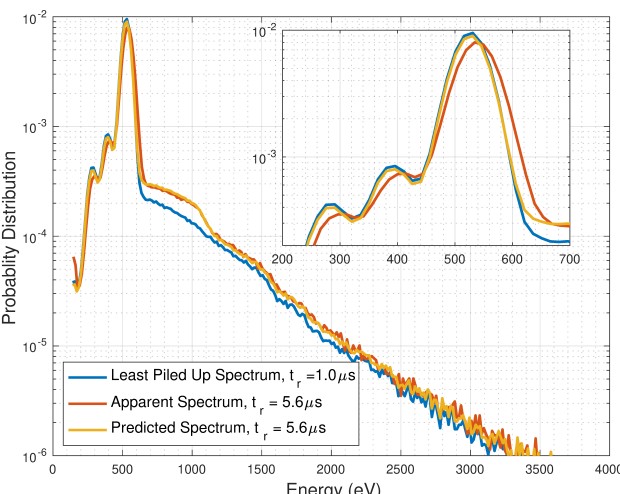

**Figure 2.** X-ray spectra from seed plasma in the PFRC-2 ($H_2$ fill-gas) formed by a continuous radiofrequency (rf) 550 W source. $t_r = 1.0$ μs (blue) and 5.6 μs (red). From the least piled-up spectrum, $t_r = 1.0$ μs, the true count rate $\mu = 9.54$ kcps is obtained. The X-ray spectrum predicted with a two-photon model for $t_r = 5.6$ μs is shown in yellow and described in Section 7. The inset shows a magnified view of the low energy region of the spectra to make clear the peak shift at high $\mu t_d$.

## 3. Pulse Pile-Up Reduction Techniques

There are numerous ways to reduce PPU and its artifacts. One is to reduce the solid angle viewed by the detector. This decreases $\mu$, hence the signal-to-noise ratio (S/N), necessitating longer data-taking time or creating larger uncertainty.

Another is to place a selective absorber to reduce the flux of X-rays in some regions of the spectrum [14]. This allows $\mu$ in the other regions of the spectrum, those of interest, to be unchanged while the total $\mu$ is decreased. This works well, though is complicated by edges in the absorber's transmission coefficient and the difficulty in finding and fabricating a thin absorber with the correct spectral features.

Other solutions are implemented *via* signal processing: (a) reducing the width of the shaped voltage pulse, a method that degrades the energy resolution; (b) rise-time discrimination of pulses; (c) tailoring the shape of the processed pulse, e.g., exponential, Gaussian, square, trapezoidal, or triangular, see Figure 3; and (d) comparing "fast" (10–200 ns) channel with "slow" (1–25 μs) channel pulse arrival times before vs. after pulse shaping, termed the fc-sc method. The limitations of these will be described shortly.

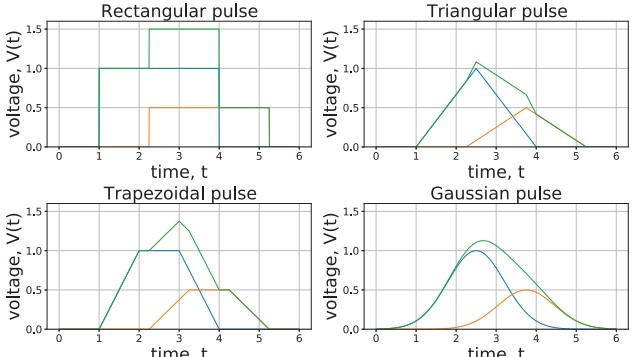

**Figure 3.** Voltage vs. time plot for two rectangular, triangular, trapezoidal, and Gaussian pulses (blue and orange) arriving within $t_d$ and them added together (green).

In previous PFRC experiments [6,7,15], the X-ray flux was low; long-duration measurements were needed. Recent experiments have produced considerably higher X-ray fluxes, providing the possibility for more detailed spectral resolution.

In many circumstances, PPU can be mitigated by using PUR techniques. In the Amptek X-123 Fast SDD's fc-sc method—typical operational parameters are listed in Appendix A—the energy of a pulse is determined using a slow channel while the arrival time of a pulse is determined using a fast channel. If the fast channel measures two pulses within $t_d$ set for the slow channel, both pulses are rejected [5]. There are circumstances in which PUR is ineffective, as when the X-ray spectrum is quasi-Maxwellian. Then the tail inferred for the spectrum may be compromised by a much brighter low energy part of the true spectrum. Moreover, the small electrical charges created by low energy photons may not be recognized by the SDD as pulses. Several low-energy photons may arrive within the 10's of ns of the fast channel's pulse-pair resolving time. This contrasts with a common use of pulse-height energy detectors that concentrate on peaks whose heights are well above a low background. Under these circumstances, better mitigation techniques are required.

## 4. Determination of Photon Energy

A photon incident on the Amptek X-123 Fast SDD generates electrons in the conduction band with a number proportional to the photon's energy. That charge is integrated by the SDD's electronics, yielding a shaped voltage pulse whose height is used to measure the photon's energy.

An ideal detector, $t_d = 0$, would register one X-ray photon of apparent energy $E$ for every X-ray photon of true energy $E_i$. When X-ray photons are incident upon an ideal detector at a spectral rate of $f_i(E_i)$ (counts/eV/second), an apparent spectral rate of $f_a(E) = f_i(E)$ occurs.

In real detectors, $t_d$ is not 0 and pulses arriving within $t_d$ create amalgamate peaks. An X-ray photon incident at time $t_i$ with energy $E_i$ produces a voltage response of a specified shape $V(t - t_i, E_i)$ whose time-integrated value is proportional to $E_i$. The voltage responses of several photons sum to a single voltage signal. Amptek SDDs use the maximum V of the summed signal, not its time-integral, to determine an X-ray's energy. If two or more pulses hit the detector with small arrival time difference, less than $t_d$, it becomes impossible to resolve the pulses by detecting the maximum of V. The pulse processor records only a single pulse and assigns an increased height, hence wrong energy, to it. This creates a false count of photons with high energy and a reduction at low energy.

The X-123 SDD-specific description of how pulse processing works is: Every photon hitting the detector results in a trapezoidal shaped voltage with selected $t_r$ and $t_f$. Pile up occurs for non-resolvable pulses whose time-dependent voltages are added to each other. The SDD pulse processor looks for peaks in the voltage signal by the following procedure: (1) If the voltage starts to rise, the PEAKH logic is set off and the logic starts to look for a peak. (2) When the voltage falls by a specified amount, the system recognizes a peak and takes it as a valid pulse. (3) The voltage at the peak is recorded as the energy of the pulse.

## 5. Modeling Pile Up for Uncorrelated Trapezoidal Pulses in the Two-Photon Approximation

Analytical results regarding pile up, in the contexts of non-paralyzable pulse detectors, have been derived by Taguchi et al. [11]. In this section, we are reviewing and re-deriving them to fit the technicalities and specifics of pulse-height silicon drift detectors which detect peaks and assign peak heights as pulse energies. These results will be used in section VII to construct our pile-up reduction algorithm.

For trapezoidal pulses, two pulses arriving with a time gap ($\Delta t$) exceeding $t_d$ have distinguishable peaks, hence are resolvable. Pile up only occurs if two pulses arrive with $\Delta t < t_d$. For $\Delta t > t_d$, the second pulse's amplitude is not offset by the first pulse as readily seen in Figure 4. (Intrinsic detector noise may be ignored, see Appendix B.). We assume no correlation exists between photon arrivals, i.e., the arrival-time probability distribution is uniformly distributed.

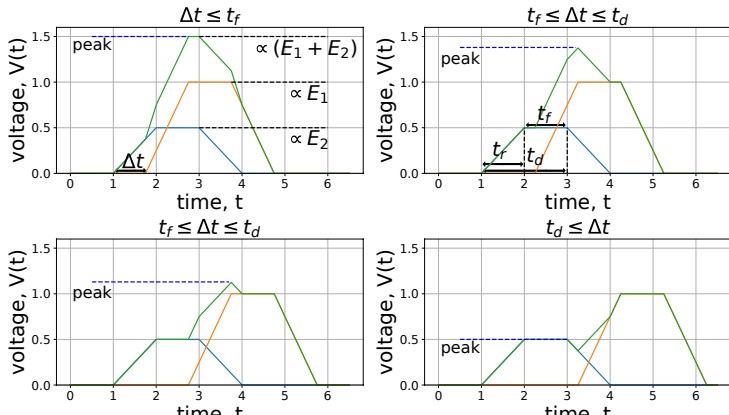

**Figure 4.** Sum (shown in green) of two trapezoidal voltage pulses (blue and orange) for different values of $\Delta t$. The sum follows different patterns and formulae for the three cases: $\Delta t < t_f$, $t_f < \Delta t < t_d$, and $t_d < \Delta t$. For $t_d < \Delta t$, the pulses are resolved and no PPU occurs.

In the peak detection modeling, two photons pile up if two sequential photons have $\Delta t$ less than $t_d$ and between the second and third photons $\Delta t$ is larger than $t_d$. After detection of a pulse, the probability of not detecting another pulse within time $t_d$ is $e^{-\mu t_d}$, equivalent to the probability of no pulse pile up being $e^{-\mu t_d}$ [16]. So the probability of detecting another pulse within time $t_d$ is $1 - e^{-\mu t_d}$. Given that arrival time is assumed to be uncorrelated, the probability of two pulses piling up is thus $e^{-\mu t_d}(1 - e^{-\mu t_d})$. We denote $1\gamma$ as a no pile-up event and $2\gamma$ as a two-photon pile-up event. For small $\mu t_d$

$$p(1\gamma) = e^{-\mu t_d} \approx 1 - \mu t_d \quad \text{and}$$
$$p(2\gamma) = e^{-\mu t_d}(1 - e^{-\mu t_d}) \approx \mu t_d. \tag{1}$$

Small $\mu t_d$ can be considered the "overlap probability," an important dimensionless parameter for estimating the amount of pile up in a spectrum.

Accordingly, the approximate apparent count rate $\mu_a$ is expressed in terms of the true count rate $\mu$. Within the apparent count events, $\mu_a p(1\gamma)$ and $\mu_a p(2\gamma)$ are the number of events where only one and two photons have arrived within $t_d$. We disregard pile-up events involving 3 or more photons. So there are, in total, $\mu_a p(1\gamma) + 2\mu_a p(2\gamma) \approx \mu$ true photons. This gives

$$\mu_a \approx \frac{\mu}{p(1\gamma) + 2p(2\gamma)} \approx \frac{\mu}{1 + \mu t_d}. \tag{2}$$

The probability of detecting an energy $E$ given two photons have piled up is $p_{2\gamma}(E)$. So the ratio of apparent pulses with energy $E$ where two photons have piled up is $p(2\gamma)p_{2\gamma}(E)$. This number gets added to the fraction of counts with energy $E$ where no pile up happened, which is $p(1\gamma)p(E)$. We can then write the ratio of apparent pulses with energy $E$, or the apparent probability spectrum, $p_a(E)$ as

$$p_a(E) \approx p(1\gamma)p(E) + p(2\gamma)p_{2\gamma}(E). \tag{3}$$

$p_{2\gamma}(E)$ needs to be expressed in terms of the energy of the first and second photons ($E_1$, $E_2$), the shape of the trapezoid, $t_d$, and $p(E)$. The shape of a trapezoidal pulse, with $t_r = t_f$, can be expressed through three parameters: $t_r$, $t_f$, and the height of the trapezoid corresponding to an energy $E$. From Figure 4, we observe that the maximum value of the peak of the summed pulse is:

$$E = \begin{cases} E_1 + E_2 & \text{if, } \Delta t \leq t_f \\ \max(E_1, E_2) \\ \quad + \min(E_1, E_2)\left(1 - \frac{\Delta t - t_f}{t_r}\right) & \text{if, } t_f \leq \Delta t \leq t_d \end{cases} \tag{4}$$

We define a shape parameter, $a \equiv t_r/t_d$, termed the triangularity: $a = 0$ corresponds to a rectangular pulse while $a = 1$ corresponds to a triangular pulse.

Because $\Delta t$ is uniformly distributed from 0 to $t_d$, the probability distribution of $\Delta t$, $p_{\Delta t}$, follows $dp_{\Delta t}/d(\Delta t) = 1/t_d$. For a uniformly distributed $\Delta t$, the probability distribution of $E$, if the energies of two photons are given, $(E_1, E_2)$, is

$$p_{2\gamma}(E|E_1, E_2)dE = \begin{cases} t_f/t_d, & \text{if } E = E_1 + E_2 \\ |\frac{dp_{\Delta t}}{d\Delta t}\frac{d\Delta t}{dE}|dE, & \text{otherwise.} \end{cases} \tag{5}$$

Equation (4) can be used to differentiate $E$ with respect to $\Delta t$ and then inverted to arrive at $|d(\Delta t)/dE| = t_r/\min(E_1, E_2)$, see Figure 5. Using the Heaviside, $\theta(x)$, and Dirac delta, $\delta(x)$, functions, Equation (5) can be written as

$$p_{2\gamma}(E|E_1, E_2) = \delta(E - E_1 - E_2)(1 - a) + \frac{\theta(E - \max(E_1, E_2))\theta(E_1 + E_2 - E)}{\min(E_1, E_2)}a. \tag{6}$$

In Figure 6, for $a = 0$, $p_{2\gamma}(E|E_1, E_2)$ is a Dirac delta function at the sum of the incident photon energies. For $a = 1$, the function is uniform between $E_1$ and $E_2$. For an intermediate $a$ value, e.g., $a = 0.5$, both features are present.

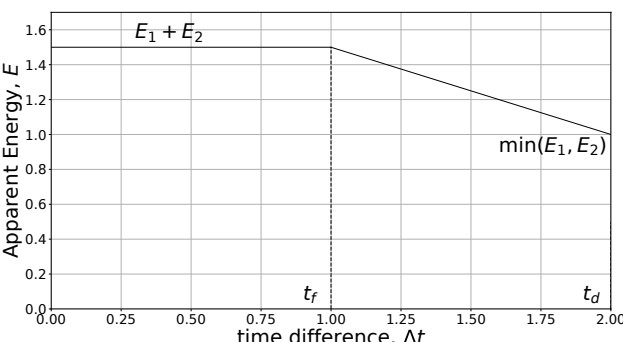

**Figure 5.** Apparent energy $E$ vs. $\Delta t$ for $0 \leq \Delta t \leq t_d$.

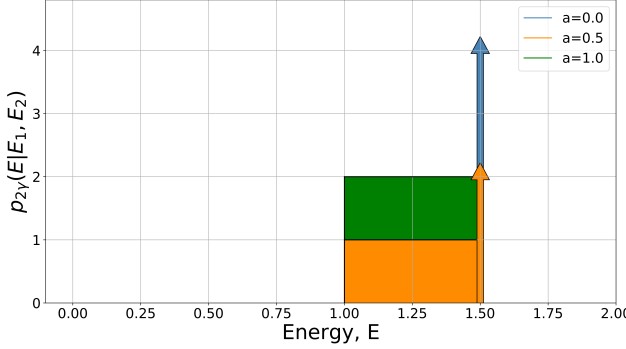

**Figure 6.** Probability density function for apparent energy, $p_{2\gamma}(E|E_1, E_2)$ vs. $E$ for several values of the triangularity $a$ with $E_1 = 0.5$, $E_2 = 1$. An upward arrow indicates a Dirac delta function at that value.

To get the energy probability distribution given two photons of any possible energy, we need to multiply $p_{2\gamma}(E|E_1, E_2)$ with the probability that photons have energy $E_1$ and $E_2$,

which is $p(E_1)p(E_2)$, and then sum over all possible cases, which is the going to be a sum over all possible combinations of $E_1$ and $E_2$. Hence, the general two-photon pile-up energy probability distribution function depends on the one-photon energy spectrum as follows:

$$p_{2\gamma}(E) = \iint p_{2\gamma}(E|E_1, E_2)p(E_1)p(E_2)dE_2dE_1. \tag{7}$$

Integration of the first term using Equation (6) yields

$$p_1(E) = \iint \delta(E - E_1 - E_2)p(E_1)p(E_2)dE_2dE_1$$
$$= \int_0^E p(E - E_1)p(E_1)dE_1. \tag{8}$$

The integration limit of $E_1$ is from 0 to $E$ because $p(E_1) = 0$ for $E_1 < 0$ as a photon's energy cannot be negative: $p(E - E_1)p(E_1) = 0$ for $E_1 > E$ or $E_1 < 0$.

In order to integrate the second term from Equation (6), notice that $p(E_1)p(E_2)\theta(E - \max(E_1, E_2))\theta(E_1 + E_2 - E)/\min(E_1, E_2)$ means that the integration is in the region with $\max(E_1, E_2) \leq E$ and $E \leq E_1 + E_2$. It is also true that the term we are integrating is symmetric with respect to $E_1$ and $E_2$. Thus the integration may be performed in the region where $E_1 \geq E_2$ and then the result is multiplied by 2. In the region $E_2 \leq E_1$, $E \leq E_2 + E_1 \leq 2E_1$ or $E/2 \leq E_1$. So $E_1$ ranges from $E/2$ to $E$. Now for a slab with constant $E_1$, $E \leq E_2 + E_1$, i.e., $E - E_1 \leq E_2$. Thus, for $E_2 \leq E_1$, $E_2$ ranges from $E - E_1$ to $E_1$ for a slab with constant $E_1$ and the integration becomes

$$p_2(E) = \iint \frac{1}{E_2}p(E_1)p(E_2)dE_2dE_1$$
$$= 2\int_{E/2}^E p(E_1) \left( \int_{E-E_1}^{E_1} \frac{p(E_2)}{E_2}dE_2 \right) dE_1. \tag{9}$$

Combining results from Equations (8) and (9) and multiplying them with necessary factors gives

$$p_{2\gamma}(E) = (1 - a) \cdot p_1(E) + a \cdot p_2(E). \tag{10}$$

These results produce the effect of a trapezoidal voltage shape function on the energy spectrum when accounting for two-photon PPU. Thus, Equation (10), applied to Equation (3), and Equation (2) give all the information about the probability distribution $p_a(E)$ and total count rate $\mu_a$ of the apparent spectrum in terms of the count rate $\mu$, the probability distribution $p(E)$ and the dead time $t_d$ of the original spectrum, in the two-photon approximation. This provides the necessary tools to analyze the pile up for trapezoidal-shaped voltage pulses in the two-photon approximation.

## 6. Ppu Examples Using the Two-Photon Uncorrelated Trapezoidal-Pulse Model

Our derived formula will be applied on both a narrow Gaussian and a truncated exponential spectra, the latter representative of EEDFs (and XEDFs) predicted for certain FRCs [13,15]. The overlap probability was taken to be $\mu t_d = 0.1$. The effects of triangularity on the spectra are first presented.

The artificially piled-up plot of a monochromatic input spectrum (approximated by a narrow Gaussian function with FWHM = 0.0526 keV) is shown in Figure 7. For pulses with $0 < a < 1$ there is the main peak, a secondary peak (at twice the energy of the monochromatic spectrum), and a constant region, roughly $1.1 < E < 1.9$ keV, in between. For $a = 1$ pulses there is no twice-energy peak. For $a = 0$ pulses there is no constant region. The constant region has a height proportional to the triangularity. The constant region is

present between channels 2600 and 4400 in the Amptek DppMCA spectrum measurements, see Figure 8 [17].

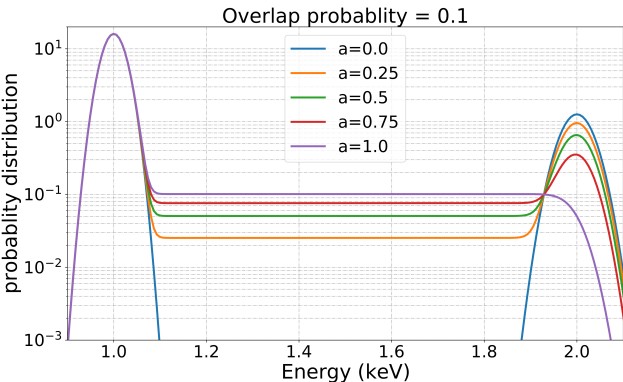

**Figure 7.** Probability distribution vs. energy of piled-up output of a narrow Gaussian for 5 different values of triangularity.

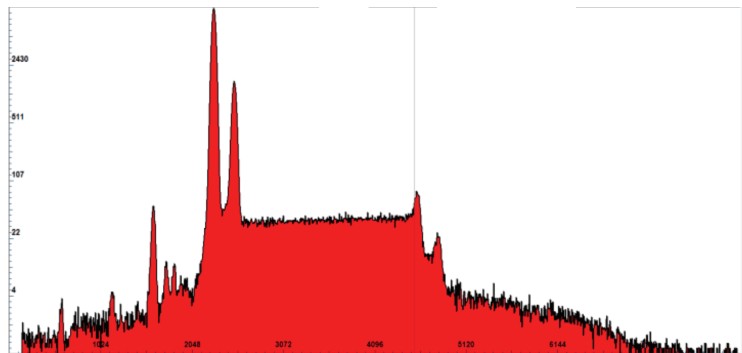

**Figure 8.** X-ray spectrum, Zn target illuminated by X-ray tube, 30 kV: 25 mm$^2$ Amptek (PUR disabled) DppMCA data [17]. Horizontal axis—energy channel number (264/keV); vertical axis—counts, log scale. Zn K-$\alpha$ at 8.6 keV, ch 2270.

The piled-up plot of an exponential function ($\propto e^{-E/E_0}$, $E_0 = 1$ keV) truncated at 1 keV is shown in Figure 9. The pile-up tail differs by more than an order of magnitude for changing triangularity. This highlights the importance of taking pulse shape into account when modeling the PPU.

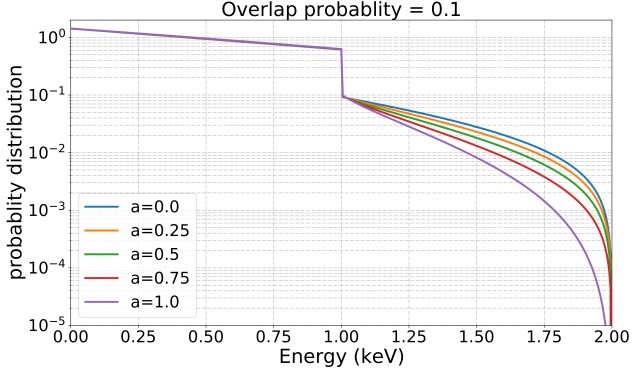

**Figure 9.** PPU spectra for truncated exponential ($\propto e^{-E/E_0}$, $E_0 = 1$ keV) spectrum with cutoff at 1 keV.

## 7. Experimental Validation of Pulse-Pile-Up Model in Two Photon Approximation

In this section the two-photon pile-up model is applied to the data collected from the graphite-target X-ray tube and 550-W rf-formed seed plasma and shows that PPU can quantitatively explain the increased-amplitude tails observed at higher CR or higher $\mu t_d$.

In the carbon target data, the pulse has $t_r = 0.2$ µs and $t_f = 0.012$ µs, hence $t_d = 0.212$ µs and $a \approx 0.943$. The exact incidence rate of the nominal 65 kcps data is $\mu = 64.8$ kcps. The overlap probability is $\mu t_d = 0.0137$, satisfying the two-photon approximation. Given that the data with 14 kcps count rate has low $\mu t_d = 0.003$, we begin by assuming that the 14 kcps data is free of pile up. We "piled up" the pile-up-less 14 kcps up using our model and compared it with the piled-up 65 kcps data. The piled-up plot for 14 kcps using our two-photon model for trapezoidally shaped pulses shows good quantitative agreement with the natural pileup from 65 kcps, see yellow curve, Figure 1. It displays an exponential shape from 5 keV to 8 keV. We conclude that most, if not all, of the tail is due to PPU. Comparison of the data and the model is summarized in Table 1.

**Table 1.** Comparison of both models with measured data from graphite target X-ray tube. The tail is assumed to be the region beyond 5 keV. See Figure 1.

| Spectra / Results | 14 kcps Spectrum | Apparent 65 kcps Spectrum | Predicted 65 kcps Spectrum |
|---|---|---|---|
| Percentage of area under Tail | 0.0222% | 0.0515% | 0.0536% |

The two-photon model was then applied to the data from seed plasma in the PFRC filled with $H_2$. In this case we assumed that the data with $t_r = 1.0$ µs is without pile up. This data was then piled it up using the model to mimic the 5.6 µs data. The observed 5.6 µs data was compared with our calculated data in Figure 2 and Table 2. As readily seen, there is good agreement in the tail region, E > 1500 eV, and the region between 650 eV and 1000 eV where the probability doubled due to PPU. The tail is "real".

**Table 2.** Comparison of both models with measured data from seed plasma in PFRC filled with $H_2$ with RMF turned off. Peaking time was 5.6 µs. The tail is assumed to be the region beyond 650 eV.

| Spectra / Results | 1 µs Spectrum | Apparent 5.6 µs Spectrum | Predicted 5.6 µs Spectrum |
|---|---|---|---|
| Percentage of area under Tail | 11.7% | 15.7% | 15.4% |

The above examples show that the two-photon PPU model with uncorrelated trapezoidal pulses provides good agreement with the observed spectra for $\mu t_d \leq 0.1$ and is particularly useful in examining whether a tail is real or a partially an artifact.

### 8. Using the Two-Photon Model to Reduce Pulse Pile-Up Tails

In the previous section we saw that the two-photon model is valid for small $\mu t_d$, an approximation used earlier when considering the 14-kcps count-rate data with 1µs peaking time. However, as seen in Figure 9, even a low but non-zero $\mu t_d$ can cause pile up in the tail region and of an amplitude that may be important to the physics. Below we describe a method to evaluate if a tail is explicable by pile up or contains a real component.

Measurement provides the apparent spectrum $f_a(E)$ which can be used to find the apparent count rate $\mu_a = \int f_a(E)dE$ and apparent probability distribution $p_a(E) = f_a(E)/\mu_a$. Firstly, reversing Equation (2) provides the real count rate,

$$\mu \approx \frac{\mu_a}{1 - \mu_a t_d}. \tag{11}$$

An iterative process then extracts $p(E)$ from $p_a(E)$. The integration of $p_{2\gamma}(E)$ goes from 0 up to $E$, i.e., photons with energy higher than the bin under inspection do not affect PPU in that bin. Using this information, Algorithm 1 was devised.

---

**Algorithm 1:** Pile-up Reduction Algorithm

---

**Data:** $p_a, \Delta E, \mu_a, t_d, a$
**Result:** $p, \mu$
$\mu = \mu_a / (1 - \mu_a t_d)$;
$i = 0$;
**while** $i < length(E) - 1$ **do**
    $p_1 = 0$;
    $j = 0$;
    **while** $j \leq i$ **do**
        $p_1 = p_1 + p[i - j]p[j]$;
    **end**
    $p_2 = 0$;
    $j = i/2$;
    **while** $j \leq i$ **do**
        $k = i - j$;
        **while** $k \leq j$ **do**
            $p_2 = p_2 + p[k]p[j]/k$;
        **end**
    **end**
    $p_{2\gamma} = (1 - a) \cdot p_1 + a \cdot p_2$;
    $p[i + 1] = (p[i + 1] - \mu t_d \cdot p_{2\gamma} \cdot \Delta E)/(1 - \mu t_d)$;
**end**
**return** $p, \mu$;

---

The algorithm first examines a low energy bin, one that has little if any pile up. That bin is used to calculate pile up in the next bin and that pile up is then subtracted. The two bins are now pile-up removed and they are used to calculate and then remove pile up in the third bin; higher energy bins require including the contributions of all bins having lower energy. By following this procedure sequentially, moving to increasingly higher bins, pile up is iteratively removed from all bins by using the pile-up-less bins that come before it.

The algorithm is first applied on the carbon target data, to explore the improbable/unphysical result that a true tail exists beyond 5 keV. As can be seen in Figure 10, the pile-up-reduced carbon target spectra for 65 kcps data, obtained by applying Algorithm 1, has reduced the tail amplitude at 6 keV by 87%. This residual tail may be caused by three (and more) photon pile up that Algorithm 1 did not remove.

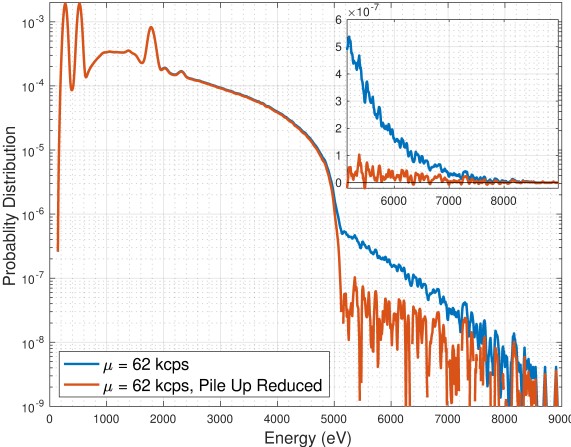

**Figure 10.** Apparent 65 kcps (blue) graphite-target X-ray-tube spectra and pile-up-reduced spectra (red) extracted using the two-photon pile-up-reduction algorithm. Noise was reduced using 50-eV-wide weighted moving-average filter. Negative values that could not be shown in logarithmic plot are seen in the linear scale inset.

To further examine this, the tail remaining in the pile-up-reduced 65 kcps data, i.e., above 5 keV, was removed altogether leaving a new spectrum. This tail-cut-off spectrum was then piled up for various count rates, see Figure 11. A key feature to notice is that the tail amplitudes for these simulated spectra are roughly proportional to the assumed $\mu t_d$. Moreover, the predicted tail for 14 kcps and 65 kcps data is in good agreement with the observed tail. It is deduced that, as $\mu t_d \rightarrow 0$, the tail would disappear.

For the seed plasma spectrum, see Figure 12, the tail in the spectra (red) with 5.6 μs peaking time remains, as does the tail in the spectra (blue) with 1.0 μs. It is evident that the pile-up-removed spectrum collapses to the original spectrum—there is no $\mu$-proportional decrease in the tail brightness. Hence we conclude that the seed-plasma tail to be of physical origin and not an artifact of pile up.

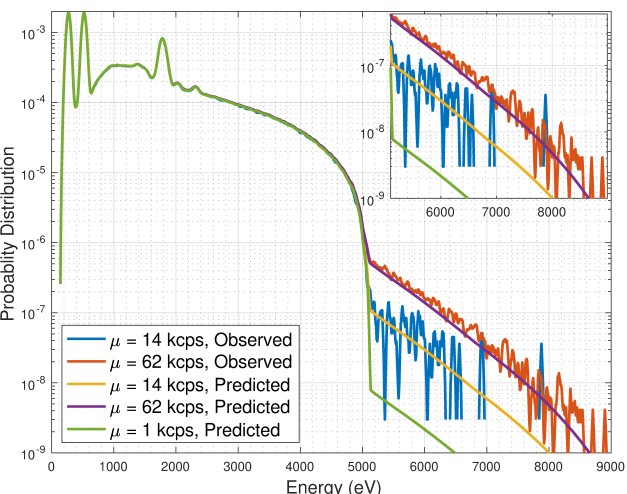

**Figure 11.** Prediction of piled up spectra using tail cut off (> 5 keV) spectrum as the original spectrum.

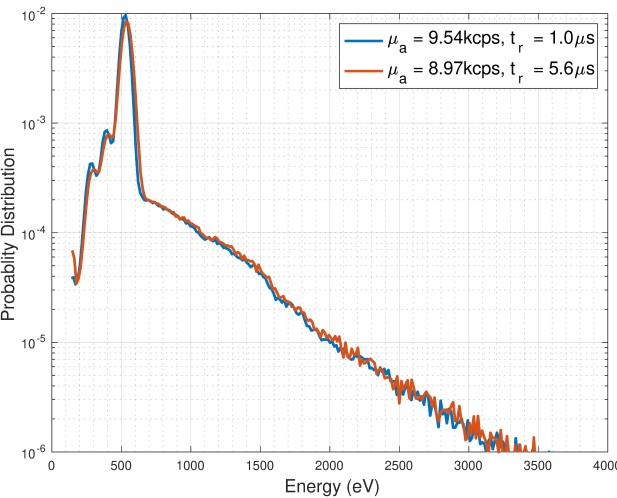

**Figure 12.** True pile-up-less X-ray spectra (with differing peaking times: 1.0 μs (blue) and 5.6 μs (red) from the seed plasma in PFRC filled with $H_2$ gas, extracted using the two-photon pile-up reduction algorithm.

## 9. Conclusions

This paper develops a two-photon model to describe changes in X-ray spectrum shapes caused by the pile-up of trapezoidal pulses in pulse-height energy spectrometers (detectors). The main focus is on the effects of PPU on the high energy continuum tail of X-ray spectra though this paper also explains changes in the low-energy region of the spectrum. The model quantifies whether *measured* high energy tails are artifacts of PPU or real. This research extends the standard use of pulse-height detectors to a new arena,

one of critical importance to warm/hot hydrogen plasma experiments where even weak high-energy electron tails, extracted from X-ray spectra, are important.

This paper quantitatively explains, for a monoenergetic X-ray source, the characteristics of a PPU-generated plateau region between the full energy and twice that energy, as a function of pulse triangularity. In the high energy part of the spectrum, the paper shows that the trapezoidal pulse shape chosen for the detector-amplifier's output plays an important role in the amplitude and shape of tails.

For spectra with $\mu t_d \leq 0.1$, the two-photon, trapezoidal, uncorrelated-pulse model accurately predicts the PPU-modified spectrum of carbon-target X-ray tube and $H_2$ seed plasma PFRC-2 data. An algorithm was developed to reduce pile up from these spectra and diagnose whether the tail regions are artifacts of PPU or have physical origins. The tail observed above the incident electron energy in the carbon-target X-ray tube spectrum was shown to be a PPU artifact.

The low energy part of the X-ray spectrum, below 200 eV—due to electronic and readout noise, and VUV photons—though far brighter than the higher energy tail, is not measured quantitatively in these experiments. Yet effects of "unresolved" low energy X-rays and noise, such as the shifting of peaks in the X-ray spectra and the brightness of this low energy photon flux, are shown in this paper and can be extracted by the two-photon model.

**Author Contributions:** T.A. formulated the 2-photon model and performed all simulations reported in this paper. C.P.S.S. assisted in performing experiments described in Figures 1 and 2 and in the analysis of this data. C.G. provided experimental data and assisted in its interpretation. S.P.V. provided the experimental data shown in Figure 2 and assisted in its interpretation. T.Q. and T.R. performed the experimental work shown in Figure 1. S.A.C. initiated and directed the research and assisted in the experimental work reported in Figures 1 and 2. All authors have read and agreed to the published version of the manuscript.

**Funding:** This work was supported by U.S. Department of Energy, Office of Science, Office of Fusion Energy Contract Number DE-AC02-09CH11466 and ARPA-E Award No. DE-AR0001099 (Princeton Fusion Systems).

**Data Availability Statement:** The digital data and unpublished material presented in this paper are available at [18]: http://arks.princeton.edu/ark:/88435/dsp01x920g025r (accessed on 30 January 2023).

**Acknowledgments:** This work was supported by U.S. Department of Energy, Office of Science, Office of Fusion Energy Contract Number DE-AC02-09CH11466, ARPA-E Award No. DE-AR0001099 (Princeton Fusion Systems), and the Princeton Program in Plasma Science and Technology. We thank R. Redus for his valuable discussions on and contributions to this paper.

**Conflicts of Interest:** The authors declare no conflict of interest.

## Appendix A

Typical operational parameters of the Amptek X-123 Fast SDD are: Clock speed 80 MHz; 0.2–25 μs peaking time; 0.12–2 μs flat top time; 34.985 total gain (fine + coarse); 204 ns detector reset lockout; 100 ns fast channel peaking time; PUR off; RTD off; MCA channels: 1024; Peak detection mode: Normal; Slow threshold: ch 22; Fast threshold: ch 14.5; BLR mode 1 (baseline restoration); BLR up/down correction 0/3; High Voltage set: −135 V; and Temperature: 240 K.

## Appendix B

The intrinsic detector noise is due to several well-known phenomena including photon statistics, Fano noise and pre-amplifier noise, the latter likely due to Johnson noise. We have measured the noise spectrum with no plasma and with detector temperatures in the range 225 to 260 K and find it to be well-described by a Gaussian centered near E = 0 and with $\sigma = 25$ eV, hence of little significance to this study.

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
