# Peer review of "Analysis and Mitigation of Pulse-Pile-Up Artifacts in Plasma Pulse-Height X-ray Spectra"

_plasma, doi:10.3390/plasma6010006_

Round 1

Reviewer 1 Report

I have read the manuscript number: plasma-2022624with title: "Analysis and mitigation of pulse-pile-up tail artifacts in warm-plasma pulse-height X-ray spectra ", and I found that authors trying to develop a two-photon model to describe changes in X-ray spectrum shapes caused by the pile-up of trapezoidal pulses in pulse-height energy spectrometers (detectors). I would like the authors to consider the following points before I recommend the publication

1-In the title, the authors have been writing warm plasma, what does the author mean by warm plasma?

2-This paper deals with a specific type of pulse that trapezoidal pulses, is there any other type of pulse? If there is another kind, what are its effects?

3- The used units for physical quantities must be reviewed.

4- What do authors mean by ( kcps)? 

5-If there is warm plasma where it is affected.

6-All references must be written in the same style and according to the type of journal.

7-The presentation and grammar need improvement. 

Author Response

We thank the referee for the effort and are grateful for the favorable review. All the suggestions made were for minor changes. We have complied with most and provide, below, responses to all the suggestions and requests.

Referee 1.

  1. The referee asks the meaning of the phrase “warm-plasma.” Though that is common jargon in the fusion community, meaning average particle energies between 50 and 500 eV, the referee is correct to questioning its undefined use in the title. We have removed it from there and added the “50-500 eV” range definition in the abstract.
  2. The referee questions our focused study of trapezoidal pulse shapes. In fact, the paper also describes, in detail, triangular and rectangular pulse shapes, which are limiting cases of trapezoidal one. Perhaps the most important reason for directing attention to the trapezoidal pulse shape is because it is the one used in modern SDDs, hence the most relevant. Another pulse shape is Gaussian. That is already included in Figures 3 and 4 and a narrow Gaussian shape is analyzed in Figure 7; additionally, we have provided a reference, 9, to a study of Gaussian pulse shapes. Another possible pulse shape is exponential, an approach used many decades ago in analog systems and now no longer in use.
  3. We have reviewed the physical units and find them correct.
  4. By kcps we mean kilo-counts per That definition has been added where kcps was first used, line 88 in the Latex document, the third paragraph in Section II, entitled “Sample Spectra with PPU.”
  5. The effect of PPU on spectra is first at the energy where the X-ray count rate is high. For our plasmas, that is at relatively low energy, below 600 eV. As explain in the paper, see figure 2, PPU causes counts to be removed from that region of the spectrum. The effect of PPU, in the two-pulse model, also occurs between that energy and twice that energy, as described in figure 7. These effects are shown in actual data, Figures 1 and 2 and in figure 9, the latter being for a truncated Maxwellian.
  6. Format of references. In the submitted version of the paper, all the journal articles had the same format, except for the page numbers and issue number. Those have been modified so all are the same. The references to websites are according to common journal format.
  7. The manuscript was primarily written by native English-speaking authors. Many small changes were made to the wording to improve the grammar. It has been reviewed by other native English-speaking scientists who judged the manuscript to be clear and grammatically correct.

Reviewer 2 Report

The authors report their work on the analysis of pulse-pile-up (PPU) behavior in photon detection, and the algorithm they used for the reconstruction of actual soft X-ray spectra out of the apparent spectra. The reconstruction algorithm is built based on a digitalized trapezoidal pulse signal, in accordance with the information provided by the detector vendor. The model was validated by comparing artificially piled-up spectra recorded at 14 kcps and naturally piled-up spectra recorded at 65 kcps, and the result shows that the reconstruction algorithm can efficiently correct the pile-up artifact. Given the fact that the manuscript provides a comprehensive explanation of their pile-up reduction algorithm, and the topic falls into the journal’s scope, the referee suggests the manuscript be published as it is.

Minor comments:

1. Page 2/8, Fig. 1 caption, it seems that ‘14 kcp (blue)’ should be ‘14 kcps (blue)’.

2. Page 4/8, ‘After detection of a pulse, the probability of not detecting another pulse within time td is e−μtd, equivalent to the probability of no pulse pile up being e−μtd.’ Please add a reference here.

3. Page 4/8, ‘In our two-photon model we disregard any more that two-photon pile-up events.’ Please check the language.

4. Reference 5 contains an invalid link. Please provide the cited content as supplementary materials or provide a new valid link. 

Author Response

We thank the referee for the effort and are grateful for the favorable review. All the suggestions made were for minor changes. We have complied with most and provide, below, responses to all the suggestions and requests.

Referee 2.

  1. The correction to the figure caption, “kcp” to “kcps,” has been made.
  2. Although we think this statement is self-evident, we have inserted a reference to a book on probability and statistics written by H.D. Brunk.
  3. That sentence has been revised to read: “We disregard pile-up events involving 3 or more photons.”
  4. That link has been updated to: https://www.amptek.com/products/x-ray-detectors/fastsdd-x-ray-detectors-for-xrf-eds/fastsdd-silicon-drift-detector